# Comparison of Leaf Area Index, Surface Temperature, and Actual Evapotranspiration Estimated Using the METRIC Model and In Situ Measurements

**DOI:** 10.3390/s19081857

**Published:** 2019-04-18

**Authors:** Arturo Reyes-González, Jeppe Kjaersgaard, Todd Trooien, David G. Reta-Sánchez, Juan I. Sánchez-Duarte, Pablo Preciado-Rangel, Manuel Fortis-Hernández

**Affiliations:** 1Instituto Nacional de Investigaciones Forestales, Agrícolas y Pecuarias (INIFAP), Blvd. José Santos Valdez No. 1200 pte. Matamoros, Coahuila 27440, Mexico; reta.david@inifap.gob.mx (D.G.R.-S.); sanchez.juan@inifap.gob.mx (J.I.S.-D.); 2Minnesota Department of Agriculture, St Paul, MN 55155, USA; jeppe.kjaersgaard@state.mn.us; 3Department of Agricultural and Biosystems Engineering, South Dakota State University, Brookings, SD 57007, USA; todd.trooien@sdstate.edu; 4Instituto Tecnológico de Torreón. Carretera Torreón-San Pedro Km 7.5, Ejido Anna Torreón, Coahuila 27170, Mexico; ppreciador@yahoo.com.mx (P.P.-R.); fortismanuel@hotmail.com (M.F.-H.)

**Keywords:** remote sensing, METRIC model, leaf area index, surface temperature, actual evapotranspiration, corn field

## Abstract

The verification of remotely sensed estimates of surface variables is essential for any remote sensing study. The objective of this study was to compare leaf area index (LAI), surface temperature (Ts), and actual evapotranspiration (ETa), estimated using the remote sensing-based METRIC model and in situ measurements collected at the satellite overpass time. The study was carried out at a commercial corn field in eastern South Dakota. Six clear-sky images from Landsat 7 and Landsat 8 (Path 29, Row 29) were processed and used for the assessment. LAI and Ts were measured in situ, and ETa was estimated using an atmometer and independent crop coefficients. The results revealed good agreement between the variables measured in situ and estimated by the METRIC model. LAI showed r^2^ = 0.76, and RMSE = 0.59 m^2^ m^−2^, the Ts comparison had an agreement of r^2^ = 0.87 and RMSE 1.24 °C, and ETa presented r^2^ = 0.89 and RMSE = 0.71 mm day^−1^.

## 1. Introduction

Crop consumptive water use varies considerably both temporally and spatially based on local and regional weather conditions and precipitation patterns, landscape location, crop type and phenological stage, soil, cropping system, irrigation, and land management. The majority of the crop consumptive use is evapotranspiration—ET. Common methods for estimating ET rely on point measurements of weather-based reference ET estimation coupled with a crop coefficient. Other methods include measurements of micrometereological fluxes in the near-surface atmospheric boundary layer at a point or along a transect [1]. Due to the heterogeneity of ET between and within fields, applying these methods results in uncertainties in the ET estimates that are not acceptable for certain applications [2]. Appropriate satellite image-based process models for estimating actual ET overcomes many shortcomings of point-based measurements by producing spatially distributed information about the available energy at the surface, including latent energy fluxes and ET [3,4,5].

The verification of remotely sensed estimates of surface variables is essential for any remote sensing study [6,7], but is becoming increasingly important with the advent of increasingly automated satellite-based image process models being applied in new geographic areas. One automated application [8] produces and publishes many of the intermediate spatially distributed products that go into the estimation of ET for the users to download for their uses—including maps of leaf area index (LAI), surface temperature (Ts), and actual evapotranspiration (ETa).

Leaf area index (LAI) is a dimensionless measure of the one-sided area of canopy foliage (m^2^) per unit ground surface area (m^2^) [9]. It is a very commonly used indicator of crop coverage and vigor. Image-based remote sensing methods estimate LAI using empirical relationships between LAI and spectral vegetation indices (VIs) [10,11] at the scale of the input imagery (e.g., 30 m for Landsat imagery). The relationships between LAI and VIs derived from satellite-estimated information have been evaluated [12,13,14] or have been estimated using other remote sensing platforms [15,16,17].

The difference between air temperature (Ta) and surface temperature (Ts) is commonly used to quantify crop water stress. For example, IRTs have been used to help refine irrigation timing and depth of application, crop yield, and quality predictions—either manually controlled [18,19,20], mounted on center pivot irrigation systems [21,22]; or mounted on unmanned aerial vehicles (UAV) [23,24,25,26] and ground-based vehicles [27,28]. Examples of estimation of Ts using satellite platforms can be found in [29,30].

A commonly used process model to estimate ET using satellite-based remote sensing is the Mapping EvapoTranspiration at high Resolution using Internalized Calibration (METRIC) model [4,31]. Studies have reported good relationships between the METRIC model and methods for ETa estimation, such as weighing lysimeter in the high desert of Idaho [4], soil water balance in the dry Texas Panhandle [32], Bowen Ratio Energy Balance Systems (BREBS) for cover crops and in vineyards [33,34], Eddy Covariance (EC) for sugar cane and riparian vegetation [35,36], and Large Aperture Scintillometer (LAS) for alfalfa [37]. A common feature of these ET estimations methods is they are advanced systems that require substantial investment, time, and skill to operate, rendering them a less viable option for practical on-farm applications. In contrast, an atmometer is an instrument that measures the amount of water evaporated to the atmosphere from a wet porous surface [38], thus mimicking the ET process from a plant. Atmometers are very useful for practical applications of on-farm water management such as irrigation scheduling [39].

Non-systematical biases in the LAI, Ts, or ETa may result in undocumented errors in the application of the output, such as water balance studies, pollution load estimations, crop water assessments, and water rights management. The purpose of this study is to make a comparison of LAI, Ts, and ETa values derived from satellite image-based remote sensing to in situ measurements in a sub-humid environment where METRIC is less commonly applied. The objectives were to: (1) Compare actual ET (ETa) estimated using an atmometer and spatially distributed ET estimates generated using the METRIC model; and (2) compare leaf area index and surface temperature, measured in situ at the time of the satellite overpass, to estimates produced by METRIC. The study was conducted for corn since it is a common and important crop in this environment. Utilizing corn will also allow for cross-comparison of the model performance to other climatic conditions where corn is also grown.

## 2. Material and Methods

### 2.1. Experimental Area

The study was carried out at a commercial corn field in eastern South Dakota (Figure 1). The corn field is located at 43° 56′ N and 96° 45′ W at 495 m above sea level. The corn row direction was north–south, the row spacing was 0.76 m, and six plants per linear meter. The final population density was 78,000 plants ha^−1^. The sources of fertilizer were beef manure or inorganic fertilizer applied preplant to achieve a yield goal of 11,300 kg ha^−1^ (180 bu acre^−1^). The field was in a corn–soybean rotation, which represented the most common cropping system in eastern South Dakota. The soil was silt loam and silt clay loam with a field capacity (FC) ranging between 0.31–0.33 m^3^ m^−3^ and a permanent wilting point (PWP) ranging between 0.15–0.19 m^3^ m^−3^, obtained using a pressure plate apparatus on soil samples collected in the field. The particle size distribution was 18% sand, 56% silt, and 26% clay, with 1–3.5% organic matter content. The 30-year average annual precipitation at the field site was 584 mm (23 inches), of which ¾ typically fell during the growing season from April to October. The 30-year mean daily maximum, minimum, and mean air temperatures were 12.3 °C, 0.3 °C, and 6.3 °C, respectively. Five observation locations were georeferenced to collect in situ measurements (Figure 1, Figure 2, and Table 1). The in situ data were collected from 2 June (day of year (DOY) 154) to 14 September (DOY 258) during the 2016 growing season.

### 2.2. Landsat Images

Six clear-sky images, collected by Landsat 7 Enhanced Thematic Mapper Plus (ETM+), Landsat 8 Operational Land Imager (OLI), and Thermal Infrared Sensor (TIRS) (Path 29, Row 29) in 2016 were used for the analysis (Table 2). The satellite images were downloaded from the United States Geological Survey (USGS) EROS Datacenter. The images were selected based on temporal coverage and cloud-free conditions. Images with clouds located more than 10 km from the area of interest were considered acceptable. The images represent top-of-atmosphere reflectance which was converted to at-surface values as described by the METRIC manual version 3.0. The wedge-shaped gaps appearing within Landsat 7 images as a result of the SLC-off issue were removed using the Imagine built-in focal analysis tool [40]. The images were processed using the METRIC model running in ERDAS Imagine [2,4,41].

### 2.3. METRIC Model

METRIC uses the following equations to estimate leaf area index, surface temperature, and actual evapotranspiration [2,42].

Leaf area index (LAI) was calculated using surface reflectance data as follows [43]:(1)$$LAI=-\frac{\ln\left[\left(0.69-SAVI\right)/0.59\right]}{0.91}.$$

For Landsat 7, soil adjusted vegetation index (SAVI) is calculated as follows:(2)$$SAVI=\frac{\left(1+L\right)\left(NIR_{band\;4\;}-\;Red_{band\;3}\right)}{L+\;\left(NIR_{band\;4}\;+\;Red_{band\;3}\right)}.$$

For Landsat 8, (SAVI) is calculated as follows:(3)$$SAVI=\frac{\left(1+L\right)\left(NIR_{band\;5\;}-\;Red_{band\;4}\right)}{L+\;\left(NIR_{band\;5}\;+\;Red_{band\;4}\right)},$$where L is a constant (L=0.1) [44].

Surface temperature (Ts) is computed using the following equation:(4)$$T_{s}=\frac{K_{2}}{\ln\left(\frac{\varepsilon_{NB\;}K_{1}}{R_{c}}+1\right)},$$where $\varepsilon_{NB\;}$is narrow band emissivity corresponding to the satellite thermal sensor wavelength band. $R_{c}$ is the corrected thermal radiance from the surface. $K_{1}$ and $K_{2}$ are constants, $K_{1}$ = 666.1 and $K_{2}$ = 1282.7 for Landsat 7 (Band 6); and $K_{1}$ = 480.9 and $K_{2}$ = 1201.1 for Landsat 8 (Band 10) [45].

Surface emissivity ($\varepsilon_{NB\;}$) is computed as follows:

For NDVI > 0:(5)$$\varepsilon_{NB\;}=\;0.97\;+\;0.0033\;\mathrm{LAI},\;\mathrm{for}\;\mathrm{LAI}\;\leq \;3;$$
(6)$$\varepsilon_{0\;}=\;0.95\;+\;0.01\;\mathrm{LAI},\;\mathrm{for}\;\mathrm{LAI}\;\leq \;3;$$
(7)$$\varepsilon_{NB\;}=\;0.98\;\mathrm{and}\;\varepsilon_{0\;}\;=\;0.98,\;\mathrm{for}\;\mathrm{LAI}\;>\;3.$$

For NDVI ≤ 0:

Water, α < 0.47, $\varepsilon_{NB\;}$ = 0.99, and $\varepsilon_{0\;}$ = 0.985.

Snow, α ≥ 0.47, $\varepsilon_{NB\;}$ = 0.99, and $\varepsilon_{0\;}$ = 0.985.

Actual evapotranspiration (ETa) was estimated using the METRIC model approach as described by [2,42]:(8)$$LE=\;R_{n}\;-\;G\;–\;H,$$where LE is the latent heat flux (W m^−2^) or ET (mm day^−1^), $R_{n}$ is the net radiation (W m^−2^), G is the soil heat flux (W m^−2^), and H is the sensible heat flux (W m^−2^).

Net radiation ($R_{n}$) is calculated using surface reflectance and surface temperature (Ts) derived by satellite imagery. Rn is the difference between incoming shortwave radiation and outgoing longwave radiation computed as:(9)$$R_{n}=\;R_{S↓\;}-\;\alpha R_{S↓}\;+\;R_{L↓}\;-\;R_{L↑}\;-\left(1\;-\;\varepsilon_{o}\right)R_{L↓},$$where $R_{S↓\;}$is the incoming shortwave radiation (W m^−2^) (solar radiation), α is surface albedo (dimensionless), $R_{L↓}\;$is the incoming longwave radiation (W m^−2^), $R_{L↑}$ is the outgoing longwave radiation (W m^−2^), and $\varepsilon_{o}\;$is the surface thermal emissivity (dimensionless). ($1\;-\;\varepsilon_{o}$) $R_{L↓}$ is the fraction of incoming longwave radiation reflected from the surface.

The input data for the Rn calculation are from weather data from the automatic Brookings weather station (i.e., surface temperature and solar radiation) and the satellite image (i.e., surface albedo and surface emissivity).

Soil heat flux (G) is the magnitude of the heat flux stored or released into the soil. G was computed using the following equations described by [44]. (10)$$\frac{G}{\mathrm{Rn}}=0.05+\;0.18{\;e}^{-0.521\;\mathrm{LAI}},\;\mathrm{LAI}\;\geq \;0.5;$$
(11)$$\frac{G}{\mathrm{Rn}}=1.80\;\left(T_{s}-273.16\right)/R_{n}+0.084\;,\;\mathrm{LAI}\;<\;0.5.$$

Sensible heat flux (H) was determined using the aerodynamic based heat transfer equation as follows:(12)$$H=\;\rho_{air}\;C_{p}\;\frac{dT}{r_{ah}},$$where $\rho_{air}\;$is the air density (kg m^−3^), $C_{p}$ is the air specific heat (1004 J kg^−1^ K^−1^), dT is the temperature difference between two heights z_1_ (0.1 m) and z_2_ (2 m), and $r_{ah}$ is the aerodynamic resistance to heat transfer (s m^−1^).

For the estimations of H, the METRIC model uses the CIMEC (Calibration using Inverse Modeling of Extreme Conditions) procedure described by [2,43]. The CIMEC approach within the METRIC model reduces possible impacts of biases in estimation of aerodynamic stability correction and surface roughness [2].

Based on LE values, the instantaneous values of ET were computed for each pixel as:(13)$$ET_{inst}=3600\;\frac{LE}{\lambda \rho w},$$where $ET_{inst}$ is the hourly instantaneous ET (mm h^−1^), 3600 is used to convert to hours, LE is the latent heat flux (W m^−2^) consumed by ET, ρw is the density of water (1000 kg m^−3^), and λ is the latent heat of evaporation (j kg^−1^)—which is computed as:(14)$$\lambda =\left(2.501-0.00236\left(T_{s}-273.15\right)\;\times \;10^{6}\right).$$

The reference ET fraction (ETrF) or crop coefficient (Kc) was calculated based on ETinst for each pixel and the reference ET (ETr) was obtained from local weather data. (15)$$ET_{r}F=\;\frac{ET_{inst}}{ET_{r}}.$$

Daily values of ET (ET24) (mm day^−1^) for each pixel were calculated as follows:(16)$$ET_{r}F\;\times \;ET_{r}24,$$where $ET_{r}F$ is the reference ET fraction, $ET_{r}24$ is the cumulative alfalfa reference for the day (mm day^−1^), and $ET_{24}$ is the actual evapotranspiration for the entire 24-hour period (mm day^−1^).

Monthly and seasonal ETa are calculated by interpolation of daily values of ETrF between images and multiplying by ETr for each day, then integrating over the specific period [2].

### 2.4. Meteorological Data

Precipitation data were collected and recorded using a tipping bucket rain gauge (TE525, Texas Instruments, Houston, TX, USA) located near observation site S. The rainfall data were reported every 30 min to be the same as the recorded soil moisture data (30 min). The remaining weather data were taken from the Brookings automated weather station located at 44° 19′ N, 96° 46′ W, 500 m above sea level approximately 40 km from the study site operated by the South Dakota Climate Office. The reference ET (ETr) was calculated using the Penman–Monteith equation [1,46]. All weather datasets were subjected to quality control (QC) using the procedures described by [1] and [46]. Hourly QC included solar radiation, air temperature, wind speed, air vapor pressure deficit, and precipitation.

### 2.5. In Situ Measurements

#### 2.5.1. Leaf Area Index Measured with AccuPAR

Leaf area index (LAI) was measured using AccuPAR model Lp-80 PAR/LAI Ceptometer (Decagon Devices, Inc. Pullman, WA, USA). The LAI measurements were collected starting 2 June (DOY 154) (vegetation stage (V3)) to 14 September (DOY 258) (reproductive stage (R6)). The probe was positioned at a 45° angle across the center row to measure PAR interception along the probe as shown in Figure 1 (II). PAR interception was measured at five geolocated locations, each location 30 m × 30 m, with five points and five replications per point above and below the corn canopy. The readings were taken between 11:00 a.m. and 12:00 p.m. every eight days, coinciding with Landsat overpass dates on clear days to minimize diffuse radiation from sky and clouds [47]. The in situ measurement of LAI obtained during the time of satellite overpass was used to assess the LAI by the METRIC model at the same pixel and the same time throughout the season. At the same time, corn height was measured on the same dates as the LAI measurements by measuring the distance between the soil surface and the tip of the longest leaf or tassel using a measuring tape. Ten plants were chosen randomly (within the pixel) for plant height measurements at each location.

#### 2.5.2. Surface Temperature Measured with Infrared Thermometer

Surface temperature (Ts) was measured with an Extech infrared thermometer model 42530 (Extech instruments Inc., Boston, MA, USA). Ts was measured every eight days from 26 June (V6, DOY 178) when the corn height was ~1.0 m, LAI = 4.5 m^2^ m^−2^, and canopy cover 80%, to 14 September (R6, DOY 258). Ts measurements were collected during cloud-free and low wind days. The infrared thermometer was held approximately 0.2 m above the corn canopy at about a 15° angle below the horizontal as shown in Figure 1 (III). The infrared thermometer had an 8:1 field of view (8 feet away the area measured is 1 foot in diameter). At each location, ten readings were taken perpendicular to the row directions, five readings pointing north and five pointing south, and then averaged. The Ts measurements of five locations were taken at the same period of time as the LAI readings (11:00 a.m. to 12:00 p.m.).

#### 2.5.3. Actual Evapotranspiration Estimated with an Atmometer

Daily actual evapotranspiration (ETa) was calculated as (17)$$ET_{a\;}=\;ET_{r}\;\times \;K_{c}$$ where alfalfa-based reference ET (ETr) was approximated using an atmometer located in Brookings, and SD and crop coefficients (Kc) from Appendix E of the ASCE Manual 70 [48] were used. Effective crop cover occurred 55 days after emergence (DAE) when the ground cover was 100% (V12). Therefore, the effective cover was used as a reference point to calculate daily $K_{c}\;$values for the growing season [48].

#### 2.5.4. Soil Moisture Measured with Soil Moisture Sensors

Soil water content was measured at three depths within the profile (0.1, 0.5, and 1.0 m) using 5TM soil moisture sensors (Decagon Devices, Inc., Pullman, WA, USA). The sensors were connected to Em50 dataloggers (Decagon Devices, Inc., Pullman, WA, USA) and measurements were recorded every 30 min during the corn growing season. The soil moisture sensors (blue triangles in Figure 1c) were installed on 30 May 2014.

### 2.6. Statistical Analysis between the METRIC Model and in situ Measurements

Linear relationships between LAI, Ts, and ETa were estimated using the METRIC model and in situ measurements were established. Statistical evaluations the of coefficient of determination ($r^{2}$) (Equation (18)), mean bias error (MBE) (Equation (19)), and root mean square error (RMSE) (Equation (20)) were computed to assess the performance of the METRIC model. (18)$$r^{2}=\;\frac{\sum_{i=1}^{n}\left(x_{i}\;-\;\bar{x}\right)\left(x_{i}\;-\;\bar{y}\right)}{\sqrt{\sum_{i=1}^{n}\left(x_{i\;}-\;\bar{x}\right)²\;\sum_{i=1}^{n}\;\left(y_{i}-\;\bar{y}\right)²\;}};$$
(19)$$MBE=\;\frac{1}{n}\;\sum_{i=1}^{n}\left(x_{i}-\;y_{i}\right);$$
(20)$$RMSE=\;\sqrt{\frac{1}{n}\;\sum_{i=1}^{n}\;{\left(x_{i}-y_{i}\right)}^{2}};$$ where n is the number of observations, $x_{i}$ is the estimated value with the METRIC model, $y_{i}$ is the measured value in situ, and the bars above the variables indicate averages.

## 3. Results and Discussion

### 3.1. Precipitation and Soil Water Content

The total precipitation during the period of study was 366 mm (21 May to 22 September 2016). Seasonal trends of soil water content (average depth) at the five locations and precipitation observed throughout the growing season are shown in Figure 3. All sites showed similar soil moisture trends during the growing season. Rainfall was evenly distributed during the growing. Low moisture levels were observed at South-East and North locations at the time of satellite overpass (Figure 3). Lower water content may be attributed to higher the higher landscape position of these two sites (Table 1 and Figure 2). The satellite overpass dates are indicated with black bars.

### 3.2. LAI Maps, Comparison and Relationship of LAI with the METRIC Model and AccuPAR

LAI maps (30 m resolution) of the entire corn field were generated as an output using the METRIC model. The LAI maps captured corn growth stages extending from the middle of the corn vegetative phase (V6, DOY 178) to late season (R6, DOY 258). Examples of the resulting maps are presented in Figure 4, with an example of high LAI values near the peak of leaf area (6.0) (R1, DOY 202) and lower LAI values as the crop senesces (2.2) (R6, DOY 258). The LAI maps developed in this study were similar to LAI maps derived from remote sensing applications by Qu et al. [2]. The similarities were that the seasonal LAI maps ranged between 1.0 and 6.0, LAI values increased from 1.0 (DOY 151) to 6.0 (DOY 192) and then decreased to 2.0 (DOY 263) at the end of the season. Colombo et al., Liang et al., and Martínez et al. [12,17,49] reported similarities in leaf area index (LAI) values for one overpass date in a corn field during three growth stages (0.8, 3.0, and 5.0).

The in situ measurements of LAI obtained during the time of satellite overpass were compared to the LAI estimates from the METRIC model. The progression and comparison of calculated and measured LAI values at the five observation locations during the 2016 growing season are shown in Figure 5 and Table 3. In the METRIC model, the maximum LAI for the entire field were found during the mid-season stage (R1-R5, DOY 202, 218, and 234) then decreased to 2.2 at the end of the growing season (R6, DOY 258). In LAI, in situ measurements at the beginning of the season were 0.27 on DOY 154 (V3). LAI values then gradually increased from 0.67 in development stage (V4) to 7.0 in mid-season stage, which occurred in the silk and kernel formation period (VT-R4, DOY 194-226). Then the LAI values decreased to 3.5 in the late season, which occurred in the physical maturity period (R6, DOY 258). The standard deviation (vertical bars) of LAI values, collected in situ with AccuPAR during the corn growing season, are presented in Figure 5.

The lowest LAI values observed at the North and South-East locations during the season may be attributed to the limited soil moisture. Those locations are at higher elevations (Table 1 and Figure 2). Limited moisture affected crop canopy development which led to lower LAI values [50]. Low moisture values also affected corn height (Figure 6). The METRIC LAI values were slightly smaller than the AccuPAR LAI values. METRIC estimates the average LAI for all plants with a 30 m × 30 m grid, whereas the AccuPAR measures the PAR interception of a few plants within a pixel (30 m × 30 m). However, both methods have errors that affect the LAI values. For example, LAI in METRIC is capped at 6 and derived from SAVI and is thus not a direct measurement (Equation 1). In AccuPAR for example, row spacing, crop height, time of measurement, and placement of the meter can affect the LAI values [51]. In our study, the placement of the sensor bar followed the manufacturer´s recommendations and the time of day was within 30 min of the satellite overpass time. In general, LAI values measured in situ with AccuPAR were greater than the LAI values estimated with the METRIC model by about 12% (Table 3).

The relationship between LAI calculated with the METRIC model; and LAI measured in situ with AccuPAR is shown in Figure 7. A good linear correlation was found between in situ measured and estimated LAI, with a coefficient of determination (r^2^) of 0.76, MBE of 0.61, and RMSE of 0.59. The large scatter at LAI around 6.0 is because the METRIC model is capped at LAI = 6, while in the AccuPAR LAI values ranged from 4.7 to 7.0. A higher coefficient of determination value (0.89) was found by Liang et al. [17]. They compared LAI measured in ground-based locations with the LICOR LAI-2000 Plant Canopy Analyzer versus LAI estimated from several vegetation indices using satellite remote sensing in different crops including corn.

The relationship and seasonal progression between average crop height and LAI measured with AccuPAR is illustrated in Figure 8. A strong relationship (r^2^ = 0.95) was found between corn plant height and LAI values until DOY 226 (R4) (Figure 8a). Similar relationship values (r^2^ = 0.99) were reported by Tasumi [44], who observed relationships between crop height and LAI for agricultural crops including corn crop in Kimberly and Idaho; and by Gao et al. [10] (r^2^ = 0.92), who took 5 to 10 representative corn plants to determine their mean height and correlated them with the LAI values. For our study, the average crop height was 0.17 m at DOY 154 (V3) and reached a maximum height of 2.2 m at around DOY 202 (R1) (Figure 8b).

### 3.3. Ts Maps, Comparison and Relationship of Surface Temperature between METRIC and Infrared Thermometer

Surface temperature (Ts) maps were derived from the METRIC model using Landsat 7 and Landsat 8 with 60 m and 100 m spatial resolution in the thermal band, respectively. Ts varied from low values (20.8 °C) to high values (29.5 °C) for DOY 258 and for DOY 202, respectively, during the growing season (Figure 9 and Table 4). Ts is affected by the water status of the plant, soil moisture content, and climatic conditions [52].

The variation of instantaneous Ts calculated with METRIC and measured in situ with the infrared thermometer in five locations is illustrated in Figure 10 and Table 4. A wide range of Ts were observed during the period of study (19–31 °C), where the coolest temperatures (~19 °C) were present at the end of the season (R6, DOY 258) and the warmer temperatures (~31 °C) were present during the mid-season (R1, DOY 202) for both methods. Slightly higher Ts values were observed at the North and South-East locations, whereas the lowest temperatures values were observed at the South, East, and East-East locations. The highest Ts values registered at the North and South-East locations were mainly due to higher elevation (Table 1) and lower soil moisture content in the root zone (Figure 3). This result is in agreement with the results reported by other researchers, e.g., Bellvert et al., Cohen et al., Anderson and Kustas, and Durigon and de Jong van Lier [24,28,29,53]. In addition, Anderson and Kustas, and Durigon and de Jong van Lier [29,53] reported that low water content in the root zone led to stomatal closure, reduced transpiration, and increased Ts.

During the growing season, the METRIC Ts values were slightly higher than infrared thermometer Ts values for the corresponding location, except on DOY 202 (Figure 10 and Table 4). However, for the whole season the METRIC values were higher than the in situ values by 0.85 °C. The difference between Ts estimated by the METRIC model is less than the accuracy of the infrared thermometer +/−2.0 °C. Some difference between the measurements should be expected, they were carried out at different scales and for different parts of the corn plants. Landsat looks down from nadir and sees canopy and some soil, while in situ measurements see the canopy only at an angle different from Landsat. METRIC reduces the potential bias in Ts calculations with the internal calibration technique CIMEC (calibration using inverse modeling at extreme conditions) [42,54]. In situ measurement biases were attributed to the time of readings (11:00 a.m. to 12:00 p.m.), assuming that readings at 11:00 a.m. are slightly colder than readings at noon. Jones and Vaughan [6] mentioned that instantaneous Ts measured in the field is very sensitive to climatic factors (e.g., cloud cover, wind speed, and solar radiation). In our study, instantaneous Ts was affected by high wind speed values and cloud cover from 1 to 2 °C and from 3 to 4 °C, respectively, less than the normal Ts values.

In our study, the standard deviation of canopy temperature (CTSD) values were less than 2.0 °C among observation locations for each date throughout the season (Figure 10). Han et al. [55] used the CTSD to classify corn water stress into three levels: Severe stress when CTSD is greater than 3.0 °C, intermediate stress when CTSD is between 2.0 and 3.0 °C, and no stress when CTSD is less than 2.0 °C. Taghvaeian et al., Zia et al., and Romano et al. [20,56,57] reported differences in canopy temperature between corn plants ranging from 2.2–3.0 °C for corn under different irrigation treatments.

The relationship between Ts estimated with the METRIC model and Ts measured in situ with infrared thermometer is presented in Figure 11. Good correlation ($r^{2}$ = 0.87) and acceptable values of MBE (0.85 °C) and RMSE (1.24 °C) were found. A similar RMSE value was reported by Neukam et al. [58], who reported RMSE less than 2.0 °C between simulated and measured canopy temperatures.

### 3.4. ETa Maps, Crop Coefficient, Comparison and Relationship of ETa between METRIC and Atmometer

Spatial ETa values were calculated with the METRIC model for the corn field. The estimated ETa values ranged between 2.7 to 9.7 mm day^−1^ (Table 5). Two ETa maps for the mid-season (DOY 194) and late season (DOY 258) are shown in Figure 12. The maps show the highest (VT, DOY 194) and the lowest (R6, DOY 258) ETa values estimated with the METRIC model during the corn growing season. ETa maps have been developed by the METRIC model on a daily, monthly, and seasonal basis from individual field scale to local scale, e.g., Liebert et al., Gowda et al., Santos et al., Choi et al., and Reyes–Gonzalez et al. a, b [36,59,60,61,62,63].

Figure 13 shows the Kc curve developed for the field, based on alfalfa-reference described by Jensen and Allen [48]. ET for the atmometer was estimated by multiplying ET measured with atmometer by the Kc in Figure 13. From the initial (V3, DOY 154) to mid-season stage (VT, DOY 194), the Kc values increased as a function of time between 30% of crop cover to 100% of effective cover, which was estimated to occur around 55 days after emergence based on field observations.

The comparison of ETa values estimated with METRIC and by the atmometer is illustrated in Figure 14. In general, the highest ETa values were found on DOY 194 (VT) (9.7 mm day^−1^ for METRIC and 8.0 mm day^−1^ for the atmometer). The smallest values were observed on DOY 258 (R6) (2.7 mm day^−1^ for METRIC and 2.3 mm day^−1^ for the atmometer).

Figure 14 shows that ETa values estimated with METRIC were greater than ETa values estimated by atmometer. However, on DOY 218 and DOY 234, the ETa values estimated with the METRIC model were lower than ETa values estimated with atmometer. The wind speed values at time of satellite overpass were low (~1.0 m s^−1^) (Figure 15). The largest difference in ETa between the METRIC model and atmometer was on DOY 194 (VT) with 1.4 mm day^−1^. The wind speed at time of satellite overpass was high (5.9 m s^−1^) (black line in Figure 15). Similar results were found by Chen and Robinson, Gleason et al., and Peterson et al. [64,65,66]. They observed that as the wind speed increased, ETr from atmometer and ETr from the Penman–Monteith (P–M) equation diverted. Chen and Robinson, and Irmak et al. [64,67] reported that the atmometer readings were relatively insensitive to wind speed.

Results of errors of daily ETa estimates for each image date between the METRIC model and the atmometer ranged between 4 to 17%. Chávez et al. [32] compared daily ETa values derived from the METRIC model and derived from soil water budget at four commercial fields, and found daily ETa estimate errors were less than 15%. Healey et al. [68] compared daily estimates of ETa from the METRIC model and from a Bowen Ratio Energy Balance System (BREBS) at three locations. They found daily ETa error around 20%. Gordillo et al. [69] compared daily ETa values calculated from the METRIC model and from eddy covariance (EC), reporting average daily ETa errors of 7%.

The relationship of ETa from METRIC and the atmometer are presented in Figure 16. In METRIC, ETa values were taken from the ETa maps, where nine average pixels of ETa around each measure point were chosen by each observation location and then averaged. The ETa values derived from atmometer are point measurements at each overpass date. The relationship showed good agreement between ETa estimations, with a coefficient of determination equal to 0.89, and MBE and RMSE equal to 0.34 and 0.71 mm day^−1^, respectively. Researchers have reported similar coefficients of determination (0.86) [36,70]. Higher coefficients (0.97) were found by Mkhwanazi et al., Gordillo et al., and Irmak et al. [37,69,71]; and lower coefficients (0.79) were reported by Healey et al. [68]. All these authors estimated daily ETa in agricultural crops using METRIC.

## 4. Conclusions

This paper compares leaf area index (LAI), surface temperature (Ts), and actual evapotranspiration (ETa) estimated by the METRIC model and in situ measurements at the time of satellite overpass over a corn field in eastern South Dakota. Comparisons based on the coefficient of determination (r^2^), mean bias error (MBE), and root mean square error (RMSE) were considered.

The outputs of LAI values from METRIC were slightly smaller (12%) than the LAI values derived from AccuPAR. This difference was attributed to the different LAI scales. However, good linear correlation was found between in situ measured and estimated LAI, with a coefficient of determination (r^2^) of 0.76 and RMSE of 0.59. In our study, the landscape position of observations was affected by soil water content, which led to low crop height, low LAI, and low canopy architecture. LAI maps derived from remote sensing can provide essential information about biomass, crop yield, and evapotranspiration at regional and local scales.

Surface temperature maps were derived with METRIC using Landsat 7 and Landsat 8 with 60 and 100 m spatial resolution in the thermal bands, respectively. For the whole season, the Ts estimated using the METRIC model was higher than the Ts measured in situ using infrared thermometer by 0.85 °C. The slight difference was attributed to the measurements, which were carried out at different scales and different parts of the plant. A good correlation (r^2^ = 0.87) and acceptable value of RMSE (1.24 °C) were found between estimated and measured Ts.

Results of comparisons between estimated ETa during the 2016 corn growing season showed that ETa values estimated with METRIC were greater than ETa values estimated by atmometer combined with crop coefficients from the literature. The largest difference in daily ETa between the METRIC model and atmometer was 1.4 mm day^−1^. This was attributed to the high wind speed values at the time of satellite overpass and the atmometer potentially underestimating the impact of wind speed. Error of daily ETa estimates for each image overpass date between the METRIC model and the atmometer ranged between 4 to 17%. The relationship revealed good agreement between ETa estimations, with a high coefficient of determination (r^2^ = 0.89) and low RMSE (0.71 mm day^−1^).

After comparing LAI, Ts, and ETa estimates using the METRIC model and in situ measurements, METRIC was found to be a useful tool to estimate these variables at a field scale in a sub-humid environment. Outputs of LAI, Ts, and ETa maps with high resolutions are key to understanding crop water stress and crop water use. Future work is needed to investigate LAI, Ts, and ETa in different crops with different irrigation systems.

## Figures and Tables

**Figure 1 sensors-19-01857-f001:**
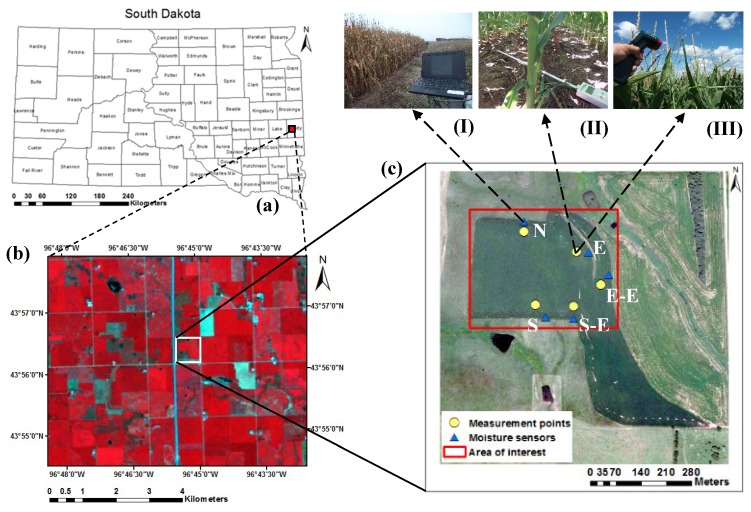
(**a**) South Dakota with county boundaries, the red rectangle shows the study area in eastern South Dakota. (**b**) Landsat 8 with false color composite (bands 5, 4, 3), the white rectangle indicates the experimental corn field. (**c**) The aerial photo with area of interest (red rectangle) shows measurement points (yellow circles) and moisture sensors (blue triangles) at the five observation locations (N, S, S-E, E, and E-E). Pictures I, II, and III show the collection of soil moisture data, LAI, and Ts, respectively.

**Figure 2 sensors-19-01857-f002:**
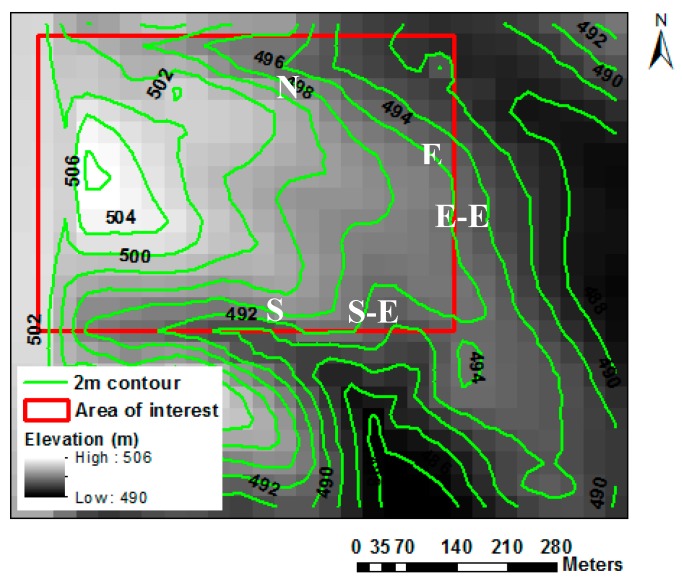
Elevation map of corn field with 2 m contour and area of interest (red rectangle), and five observation locations (white letters).

**Figure 3 sensors-19-01857-f003:**
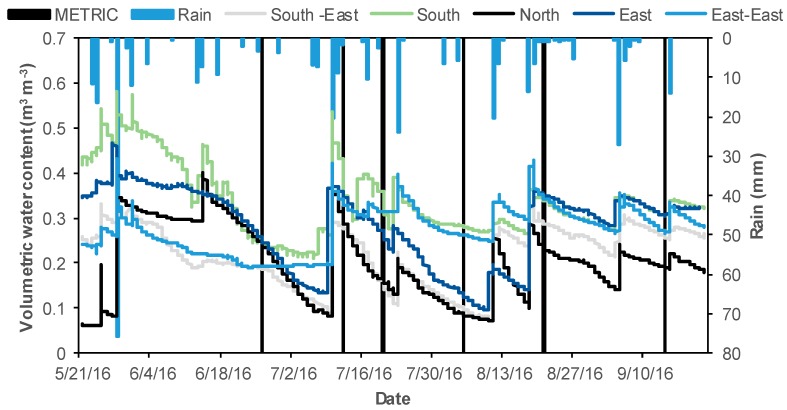
Seasonal trends of soil water content (average depths) at the five observation locations. The blue bars indicate precipitation throughout the corn growing season and the black bars denote remote sensing overpass dates (METRIC).

**Figure 4 sensors-19-01857-f004:**
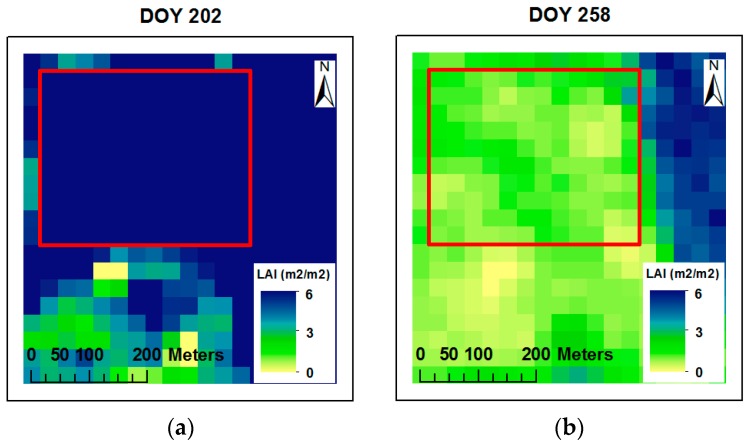
Spatial and temporal LAI maps developed from the METRIC model for two overpass dates ((**a**) DOY 202 and (**b**) DOY 258). The red rectangle indicates the area of interest within the corn field.

**Figure 5 sensors-19-01857-f005:**
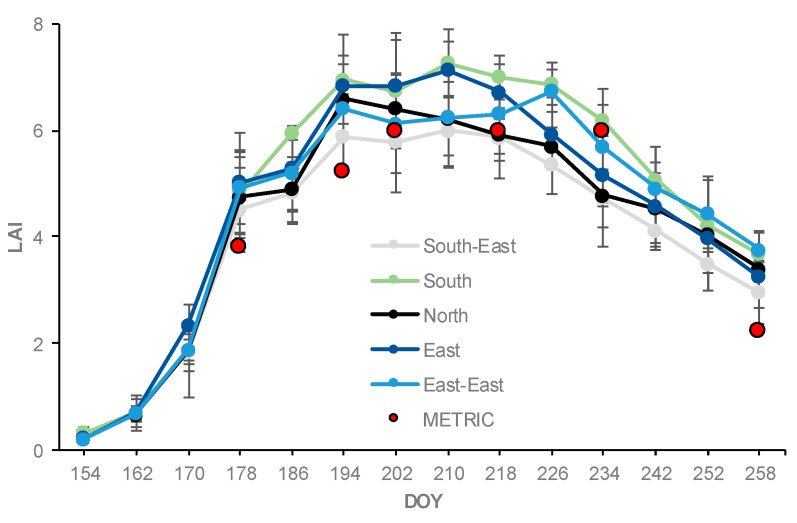
Seasonal progression and comparison of LAI estimated with the METRIC model (average of the five locations in each date, red circles) and measured with AccuPAR (five locations, each location with five points and five replications per location) throughout the season. Vertical bars represent 2 standard deviations of LAI values measured in situ with AccuPAR.

**Figure 6 sensors-19-01857-f006:**
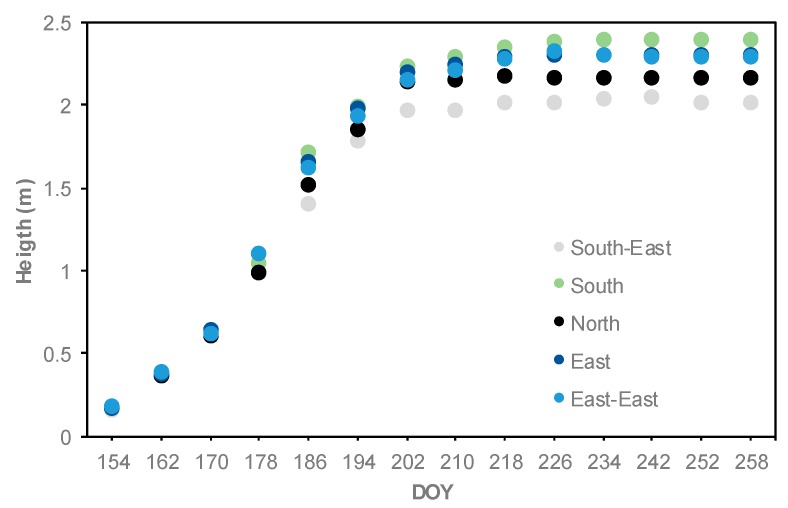
Seasonal progression of corn height at five observation locations throughout the 2016 growing season.

**Figure 7 sensors-19-01857-f007:**
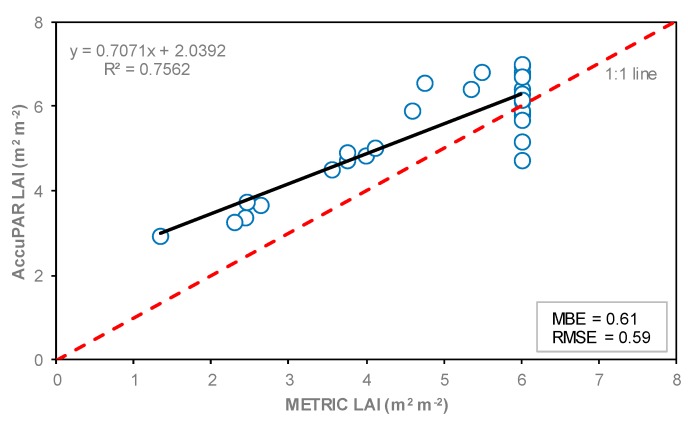
Relationship between LAI values estimated with the METRIC model and LAI values measured with AccuPAR in five observation locations during the 2016 corn growing season. The red dashed line represents the 1:1 line.

**Figure 8 sensors-19-01857-f008:**
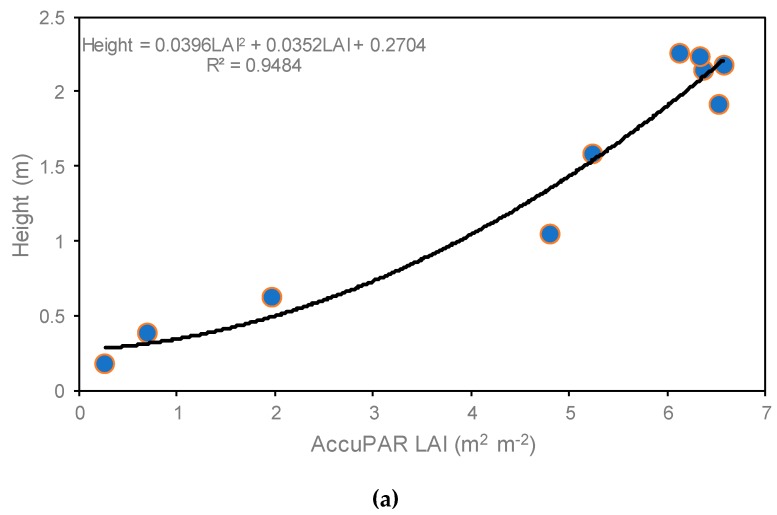
(**a**) Relationship between average crop height and average LAI. (**b**) Seasonal progression of crop height (ten reading average for each location) and LAI measured with AccuPAR (average of five locations, each location with five points and five replications per location) throughout the 2016 growing season.

**Figure 9 sensors-19-01857-f009:**
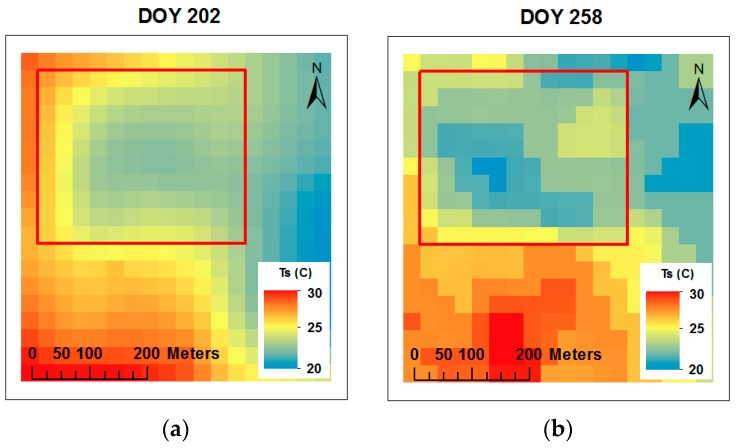
Maps of land surface temperature derived from the METRIC model, acquired using Landsat 7 (**b**) DOY 258) and Landsat 8 ((**a**) DOY 202) with 60 m and 100 m spatial resolution, respectively, throughout the 2016 corn growing season. The red rectangle indicates the area of interest within corn field.

**Figure 10 sensors-19-01857-f010:**
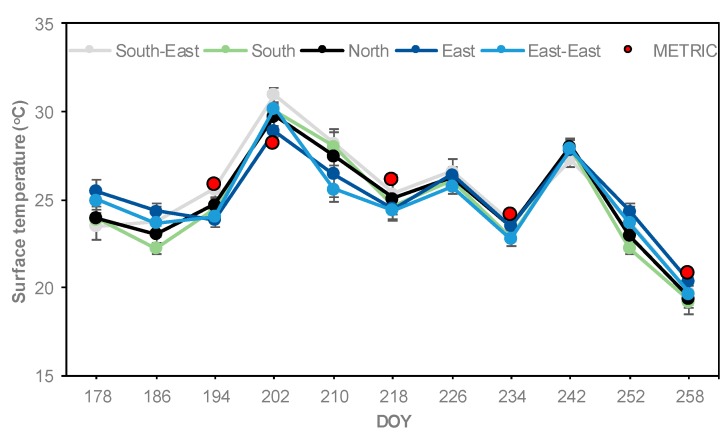
Seasonal comparison of instantaneous surface temperature (Ts) calculated with the METRIC model—red circles (values at the time of satellite overpass date for corresponding location)—and measured in situ with infrared thermometer (ten readings average in each location). Vertical bars represent standard deviations of Ts values measured with infrared thermometer.

**Figure 11 sensors-19-01857-f011:**
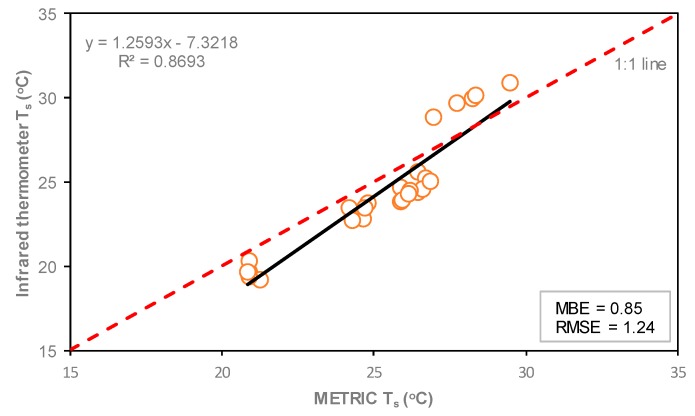
Linear correlation of Ts between the METRIC model and infrared thermometer of corn throughout the 2016 growing season. The red dashed line represents the 1:1 line.

**Figure 12 sensors-19-01857-f012:**
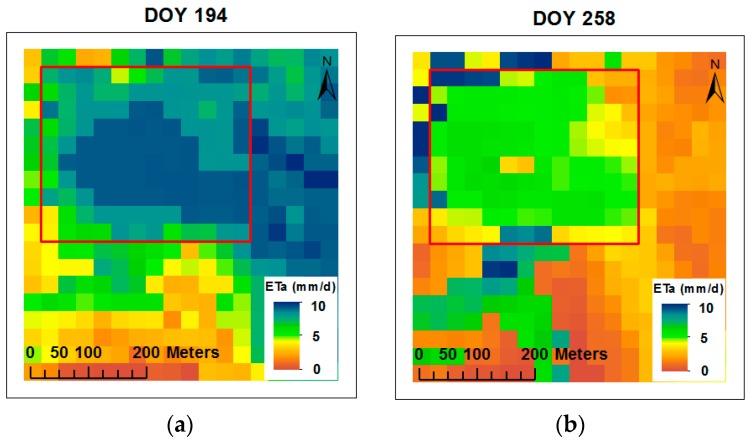
Actual evapotranspiration (ETa) maps developed by the METRIC model for mid-season ((**a**) DOY 194) and late season (**b**) DOY 258) during the 2016 corn growing season. The red rectangle indicates the area of interest within the corn field.

**Figure 13 sensors-19-01857-f013:**
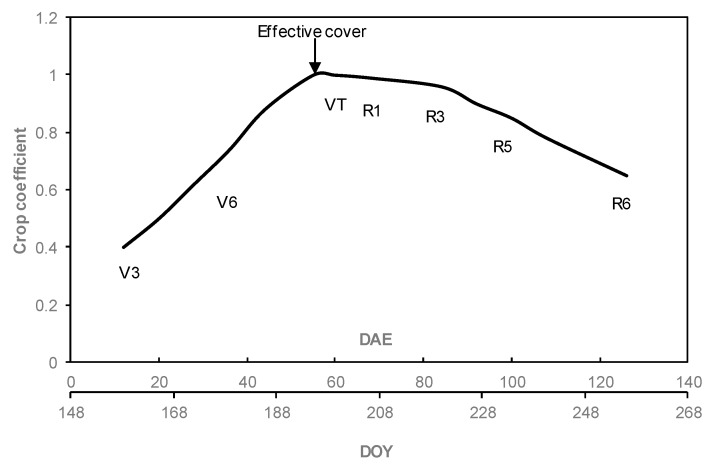
Crop coefficient curve of corn field throughout the 2016 growing season. DAE is days after emergence, DOY is day of year, and “V” and “R” refer to crop development stages.

**Figure 14 sensors-19-01857-f014:**
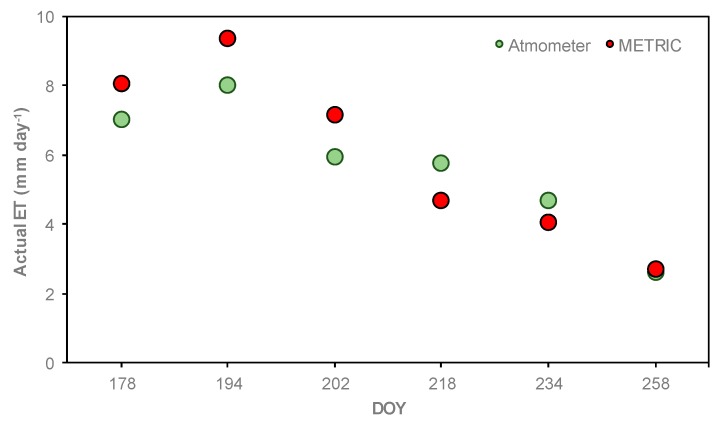
Progression and comparison between daily ETa estimated by the METRIC model (average values of each overpass date) and estimated by atmometer and crop coefficients (Kc) during the 2016 corn growing season.

**Figure 15 sensors-19-01857-f015:**
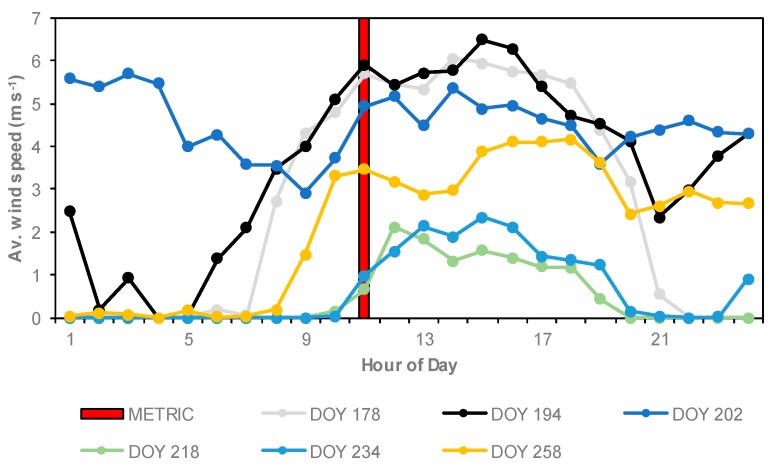
Hourly average wind speed values at the Brookings weather station for days at the satellite overpass. The red column represents the time of satellite overpass (METRIC) (~11:12 a.m.).

**Figure 16 sensors-19-01857-f016:**
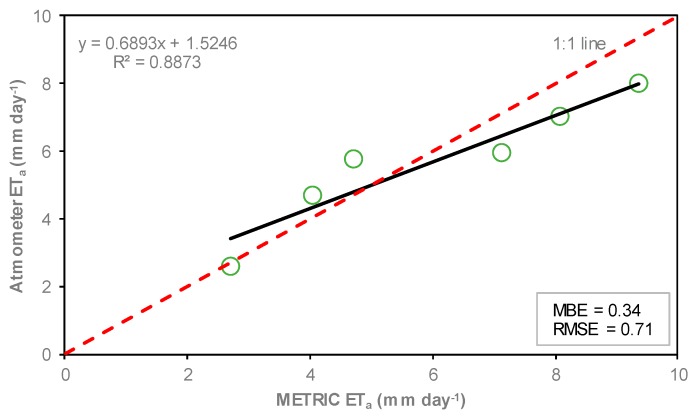
Relationship between ET_a_ estimated by the METRIC model and estimated by atmometer during the period of study. The red dashed line represents 1:1 line.

**Table 1 sensors-19-01857-t001:** Observation locations, altitude, soil texture, field capacity (FC), and permanent wilting point (PWP).

Site	Latitude (N)	Longitude (W)	Altitude (m)	Soil Texture	FC (m^3^ m^−3^)	WP (m^3^ m^−3^)
South-east (S-E)	43° 56′ 20.7″	96° 45′ 11.6″	495	silt clay loam	0.33	0.19
South (S)	43° 56′ 20.8″	96° 45′ 15.7″	493	silt loam	0.31	0.15
North (N)	43° 56′ 27.6″	96° 45′ 19.5″	501	silt clay loam	0.33	0.19
East (E)	43° 56′ 25.6″	96° 45′ 11.5″	493	silt loam	0.31	0.15
East-east (E-E)	43° 56′ 23.0″	96° 45′ 10.0″	492	silt loam	0.31	0.15

**Table 2 sensors-19-01857-t002:** DOY, acquisition dates, satellite platform, path/row, and overpass time of the imagery used for the 2016 growing season.

DOY	Acquisition Dates	Satellite	Path/Row	Overpass Time (Local)
178	06/26/16	Landsat 7	29/29	11:13:56 a.m.
194	07/12/16	Landsat 7	29/29	11:13:55 a.m.
202	07/20/16	Landsat 8	29/29	11:11:21 a.m.
218	08/05/16	Landsat 8	29/29	11:11:24 a.m.
234	08/21/16	Landsat 8	29/29	11:11:30 a.m.
258	09/14/16	Landsat 7	29/29	11:14:05 a.m.

**Table 3 sensors-19-01857-t003:** Comparison of LAI values estimated with the METRIC (MET) model and measured with AccuPAR (Accup) at five locations during the 2016 growing season, for assessing the LAI by the METRIC model.

	LAI (m^2^ m^−2^)									
DOY	South-East		South		North		East		East-East	
	MET	AccuP	MET	AccuP	MET	AccuP	MET	AccuP	MET	AccuP
178	3.6	4.5	4.0	4.8	3.8	4.7	4.1	5.0	3.8	4.9
194	4.6	5.9	6.0	7.0	4.7	6.6	5.5	6.8	5.3	6.4
202	6.0	5.8	6.0	6.7	6.0	6.4	6.0	6.8	6.0	6.1
218	6.0	5.9	6.0	7.0	6.0	5.9	6.0	6.7	6.0	6.3
234	6.0	4.7	6.0	6.2	6.0	5.7	6.0	5.2	6.0	5.7
258	1.3	3.0	2.6	3.7	2.4	3.4	2.3	3.3	2.5	3.8

**Table 4 sensors-19-01857-t004:** Ts values from the METRIC (MET) model and infrared thermometer (IRT) at five locations and five dates during the 2016 corn growing season.

	T_s_ (^°^C)									
	South-East		South		North		East		East-East	
DOY	MET	IRT	MET	IRT	MET	IRT	MET	IRT	MET	IRT
194	26.4	25.6	26.4	24.5	25.9	24.7	25.9	23.9	25.9	24.0
202	29.5	31.0	28.2	30.1	27.7	29.7	26.9	28.9	28.3	30.2
218	26.7	25.3	26.6	24.6	26.9	25.1	26.2	24.5	26.1	24.4
234	24.8	23.8	24.6	22.8	24.7	23.5	24.2	23.5	24.2	22.8
258	20.9	19.7	21.2	19.2	20.9	19.4	20.9	20.4	20.8	19.7

**Table 5 sensors-19-01857-t005:** ETa values estimated by the METRIC model for five observation locations and six overpass dates during season.

	METRIC ET_a_ (mm day^−1^)					
DOY	South-East	South	North	East	East-East	Average
178	7.98	8.06	7.68	8.45	8.22	8.08
194	9.40	9.41	8.87	9.72	9.45	9.37
202	7.16	7.08	7.10	7.14	7.26	7.15
218	4.63	4.63	4.56	4.82	4.92	4.71
234	3.91	3.96	3.93	4.23	4.23	4.05
258	2.69	2.87	2.75	2.60	2.67	2.72

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
