# Peer review of "Comparison of Leaf Area Index, Surface Temperature, and Actual Evapotranspiration Estimated Using the METRIC Model and In Situ Measurements"

_sensors, 2019, doi:10.3390/s19081857_

Reviewer 1 Report

The paper compared LAI, Ts and ETa empirically estimated from remote sensing data with field measured data. I appreciate the efforts the authors spent on the paper. However, I feel the study is more like a classroom exercise and cannot recommend it for publication in a scientific journal. The main issue with the paper is its lack of innovation. There are many similar studies. Authors need to think deeply what’s the contribution to our current knowledge of LAI, Ts and ETa estimation and validation. L122 “METRIC uses physically based equations to estimate leaf area index, surface temperature and actual evapotranspiration”. I won’t call the method in Section 2.3 “physically based equations”. They are all empirical methods to me. Section 2.5.1 The AccuPAR method for LAI has some limitations, e.g. Fang et al. (2014, AFM). Fang, H., Li, W., Wei, S., and C. Jiang, 2014. Seasonal variation of leaf area index (LAI) over paddy rice fields in NE China: Intercomparison of destructive sampling, LAI-2200, digital hemispherical photography (DHP), and AccuPAR methods. Agricultural and Forest Meteorology, 198-199(0): 126-141, doi: 10.1016/j.agrformet.2014.08.005. Section 3.2 I didn’t get any new information from the comparison of LAI. L399-402. It’s easy to list many references. How are these other studies related to your work? L431-438 same issue. Just a list of references.

Author Response

Response to Reviewer 1 Comments

Point 1: L22 METRIC uses physically based equations to estimate leaf area index, Surface temperature and actual evapotranspiration. I won´t call the method is Section 2.3 “phisically based equation. They are all empirical methods to me.

Response 1: Thank you for this comment: We agree some of the equations are empirical while others are physical or derived from physical principles. To avoid confusion, we have reworded the sentence as shown below:

Please refer to the attached pdf file which contains the pictures indicates the revision.

Point 2: Section 2.5.1. The AccuPAR for LAI has some limitations, e.g. Fang et al (2014, AFM). Fang, H., Li, W., Wei, S., and C. Jiang, 2014. Seasonal variation of leaf área index (LAI) over paddy rice fields in NE China: intercomparisons of destructive sampling. LAI-2200, digital hemispherical photography (DHP), and AccuPAR methods. Agricultural and Forest Metereology, 198-199 (0): 126-141, doi:10.1016/j.agrformet.2014.08.005.

Response 2: We appreciate the reviewer pointing this out, and further note that row spacing, crop height, time of measurement and placement of the meter can affect the LAI values (Tewolde et al, 2005). Also, Garrigues et al, 2008 and Fang et al., 2014, compared leaf area index retrievals from LAI-200, digital hemispherical photography (DHP), and AccuPAR over croplands. They concluded that the large discrepancies between retrievals techniques were observed over short canopies such as soybean, alfalfa and paddy rice, respectively, due to instruments placed below short canopies may  disturb canopy structure affecting measurments. For that reason better agreement is found for tall canopies like corn. To minimize these errors we carefully followed the recommendations by the manufacturer and suggestions in the literatura when making the measurements.

Please refer to the attached pdf file which contains the pictures indicates the revision.

Point 3: Section 3.2.1. didn´t get any new information from the comparison of LAI.

Response 3: Thank you for this comment, we assume the reviewer is referring to Figures 4 and 5. Our intent with showing these figures is to provide the reader with information about the uniformity in LAI across the field as well as show the alignment of LAI estimates between AccuPar and METRIC. We appreciate that this information may represent a notion of “presenting too much information”. At this time we have retained Figures 4-5 in the manuscript since we think this information has value and it is not entirely clear to us if the reviewer is requesting us to remove them and/or condense the text. We are very open to discuss this issue further, if desired.

Point 4: L399-402. It´s easy to list many references. How are these other studies related to your work?

Response 4: Corrected: We have reviewed our references and retained the ones that most closely relate to teh work in this study, including studies that use the METRIC model to generate ET maps, as shown below.

 Please refer to the attached pdf file which contains the pictures indicates the revision.

Point 5: L431-438 same issue. Just a list of reference.

Response 5: We apprecaite this comment: In this case, we find the studies we are referencing(Chavez et al. 2007, Healey et al. 2011, and Gordillo et al. 2014)are all relevant to our work and represent useful quantifications of discrepancies in ETa estimates between METRIC and other, independent estimates of ETa. Therefore, most respectfully, we do not find these references are ‘just a list of references’, but instead represent valuable and relevant information that helps frame our discussion or the results from our study. We are very open to discuss this issue further, if needed.

Please refer to the attached pdf file which contains the pictures indicates the revision.

Reviewer 2 Report

see attached file

Author Response

Response to Reviewer 2 Comments

Landsat images

Point 1: Remotely-sensed inputs are 6 Landsat images. Nothing is said about pre-processing of images: it is needed to specify if used data are at-satellite reflectance (Top of Atmosphere) or ground reflectances (Top of canopy); in the latter case, which atmospheric correction has been applied?.

Response 1: We appreciate this comment: Thank you for this comment. We have added the following information to the manuscript to clarify this:

Please refer to the attached pdf file which contains the pictures indicates the revision.

SAVI-LAI relationship

Point 2: Equation 1 is erroneous and should be written as follows:

LAI = -

The reference to this equation is not Bastiaanssen 1998a (Ref. 39) but Bastiaanssen 1998 b (Bastiaanssen 1998a paper is the formulation of SEBAL model, where NDVI has been selected to describe the general effect of vegetation on Surface fluxes). This formulation originates from mean values of coefficients C1, C2 and C3 (respectively 0.69, 0.59 and 0.91) obtained from 11 crops or vegetation types (Bastiaanssen, 1998b)

Response 2: Corrected: we change the equation and we add the reference (Bastiaanssen 1998b) in the references section

Please refer to the attached pdf file which contains the pictures indicates the revision.

Point 3: On my question if the means values are the best choice when working with a specific crop (corn). In situ spectral measurements would be required, in addition to LAI measurements, to answer this question. Also LAI is limited to value = 6 in this formulation but LAI may reach 7 in the studied case (mid- season stage).

Response 3: Thank you for this comment: We agree that different crops exhibits different characteristics relative the relationship between LAI and SAVI. In this study however, our purpose was to apply METRIC in its published and commonly used form in order to evaluate that model’s performance. Exploring other formulations of the relationship between LAI and SAVI is good, but it falls outside the scope of this study. In METRIC, the LAI values are derived from SAVI. Maximum values for LAI of 6.0 corresponding to SAVI = 0.69. Beyond SAVI = 0.69 the value from LAI “saturates” and does not change significantly. In METRIC, LAI is limited to 6.0 when SAVI >0.687 and LAI = 0 when SAVI<0.1. (Allen et al. 2007).

 Please refer to the attached pdf file which contains the pictures indicates the revision.

Point 4: An other remark is about the L=0.1 values in the SAVI formulation. This value was suggested by Tasumi, 2003 (Ref. 40) and is suited for very dense crops (in this case, SAVI is close to NDVI). Applying L=0.1 values to early stages of crop development is not appropriate and may explain the overestimation of METRIC derived LAI vs measured Accupar LAI (Figure 7). The use of L=0.5 value has been adopted in most studies; an example for wheat and corn seasonal crop development monitoring is given by padilla (2011).

Response 4: We appreciate this comment: A value of 0.5 frequently appears in the literature for L (Huete, 1988). For METRIC applications in the western U.S., we use L = 0.1, based on studies by Tasumi et al. (2003).

 Please refer to the attached pdf file which contains the pictures indicates the revision.

Point 5: Another solutions would be to use an adjustable L value, as it is the case in the MSAVI formulation (Qi et al., 1994):

MSAVI1 = [(NIR-Red)/(NIR+Red+L)]*(1+L)

Whit L01-2*s*(NIR-Red)*(NIR-s*Red)/(NIR+Red), but it will be needed to evaluate s, the slope of the soil line from a plot of red versus near infrared brightness values.

Studies using radiation Transfer Models have found that MSAVI was the most sensitive Vegetation Index to LAI (Brone and Leblanj, 200). Wu et al. (2007) concluded from their study on corn and potato canopies: “For field scale agricultural applications, MSAVI-LAI relationships are easy-to-apply and reasonably accurate for estimating LAI” The alternative use of MSAVI2 (qi et al, 1994) can be suggested, because it is not requiring an adaptation factor:

Response 5: We appreciate this comment. As mentioned above, the purpose of this study was to apply METRIC in its standard and published form. These suggestions provided by the reviewer would help formulate the objectives of future studies exploring sensitivities within METRIC and evaluate other formulations of the calculation routines of the model.

Please refer to the attached pdf file which contains the pictures indicates the revision.

Surface temperature equation:

Point 6: How is calculated emissivity parameter?

Response 6: Thank you for this comment: The Surface emissivity (Ɛ) is computed in model F04 using the following empirical equation developed by Tasumi et al. (2003) based on soil and vegetation thermal spectral emissivities from the MODIS UCSB emissivity Library. NDVI is used to filter soil and vegetation (NDVI>0) from water and snow (NDVI≤0).

 Please refer to the attached pdf file which contains the pictures indicates the revision.

Point 7: The coefficients K1 and K2 are given for use with Landsat thermal channel radiances. The references to K1 and K2 values is inadequate (Ref. 41: Landsat technical note from 1986, and Landsat 7 launched in 1999, and Landsat 8 in 2013. Coefficients given for Landsat 8 are the same as cited in Kamran et al. (2015) but these authors are using TIRS channel 11 instead of channel 10 as in the present paper. Please give information and references about K1 and K2 values for Landsat 7 channel 6 and Landsat 8 channels 10 y 11.

Response 7: We appreciate this comment: We have updated this reference to Chander et al, 2009. Also we change the reference in the manuscript. The values of K1 and K2 for Landsat 7 are showed in the table below and are the same that we used in our calculations. For Landsat 8 we used we used the coefficients supplied with the Landsat 8 metadata (Table 4b below).

Please refer to the attached pdf file which contains the pictures indicates the revision.

Point 8: Furthermore, in the case of Landsat 8 the use of both channels and Split-window algorithm (cf. Kamran et al., 2015) could give more accurate estimation of Surface temperature and improve the results obtained at Figure 11 (underestimation for low temperatures)

Response 8: Thank you for this comment: According to the USGS Landsat Level-1 data product, “it is not recommended that the Band 11 be used for the split-window technique”. (See Section A1 of the Landsat 8 Data Users Handbook). Therefore, we did not use that approach. Also, in this study the low temperatures in Figure 11 and Table 4, occurred when we used Landsat 7 (e.g. DOY 258 (20.8°C)).

 Please refer to the attached pdf file which contains the pictures indicates the revision.

Reviewer 3 Report

This paper (sensors-452809) provides a thorough evaluation of METRIC model derived Leaf Area Index (LAI), surface temperature (Ts), and evapotranspiration (ET) against in-situ measurements over a maize field. The manuscript is well-written, and the experiments were very well designed. LAI, Ts, and ET were estimated using the METRIC model using six Landsat images. In situ measurements were carefully made at the time of satellite overpass. The paper thoroughly analyzed the agreement between simulated and ground measured variables and carefully discussed any deviation. Below are my specific comments and suggestions.

Title: The title is a little confusing. The word “relationship” implies some connection between LAI, Ts, and ET.

Introduction:

Line 57 – 59: These two sentences seem redundant. Please try to simplify. 

Line 66 – 68: Please rephrase. This is not a valid hypothesis. Validation again in situ measurements is a way of showing whether a model is valid or appropriate for the settings. 

Line 71: Again, the word “relationship” could be misleading here.

Materials and Methods:

Line 83 – 84: Can you provide information on how the FC and PWP are obtained?

Line 108 – 110: Please provide the specific Landsat image product level (i.e. TOA or surface reflectance?).

Line 142 – 149: Can you give specific sources (i.e. weather records, surface reflectance, Ts) of each component of Rn?

Line 313: It might be interesting to show another scatter plot in Figure 7, that uses the empirical LAI-VI relationship (Eq(1)) without the cap at LAI = 6.

Figure 8: Please slightly enlarge the font.

Conclusions:

Line 464: The differences in the relationship between LAI and vegetation indices can vary due to reasons beyond spatial scales, such as crop type, structure, phenology, etc. Please refer to (Nguy-Robertson et al, 2012, Agronomy Journal, DOI: https://doi.org/10.2134/agronj2012.0065; Kang et al., 2016, Remote Sensing, DOI: https://doi.org/10.3390/rs8070597) and provide a deeper discussion.

Line 482: Again, “relationship between LAI, Ts, and ETa” is misleading.

Author Response

Response to Reviewer 3 Comments

Point 1: Title: The Title is a little confusing. The word “relationship” implies some connections between LAI, Ts and ET.

Response 1: Yes, good point. We have revised the title as follows: Comparison of Leaf Area Index, Surface Temperature, and Actual Evapotranspiration Estimated using the METRIC model and in-situ Measurements

Please refer to the attached pdf file which contains the pictures indicates the revision.

Introduction:

Point 2: Line 57-59: These two sentences seem redundant. Please try to simplify.

Response 2: Thank you for this comment, We have reduced this to one sentence.

Please refer to the attached pdf file which contains the pictures indicates the revision.

Point 3: Line 66-68: Please rephrase. This is not a valid hypothesis. Validation again in situ measurements is a way of showing whether a model is valid or appropriate for the settings.

Response 3: Corrected: The hypothesis was rephrased.

Please refer to the attached pdf file which contains the pictures indicates the revision.

Point 4: Line 71: Again, the word “relationship” could be misleading here.

Response 4: Corrected: We have revised the sentence as shown below.

Please refer to the attached pdf file which contains the pictures indicates the revision.

Materials and methods:

Point 5: Line 83-84: Can you provide information on how the FC and PWP are obtained?

Response 5: We appreciate this comment: The FC and PWP were obtained in lab using a pressure plate apparatus. The pressure were 0.33 bar for FC and 15 bars for PWP.

Please refer to the attached pdf file which contains the pictures indicates the revision.

Point 6: Line 108-110: Please provide the specific Landsat image product level (i.e. TOA or Surface reflectance?)

Response 6: We appreciate this comment: Thank you for this comment. We have added the following information to the manuscript to clarify this:

Please refer to the attached pdf file which contains the pictures indicates the revision.

Point 7: Line 142-149: Can you give specific sources (i.e. weather records, Surface reflectance, Ts) of each component of Rn?

Response 7: We appreciate this comment, we add some inputs data for the Rn calculations, for instance solar radiation comes from weather data and albedo comes from satellite image. 

Please refer to the attached pdf file which contains the pictures indicates the revision.

Point 8: Line 313: It might be interesting to show another scatter plot in Figure 7, that uses the empirical LAi-IV relationship (Eq(1)) withouth the cap at LAI=6

Response 8: Thank you for this comment: We agree that different crops exhibits different characteristics relative the relationship between LAI and SAVI. In this study however, our purpose was to apply METRIC in its published and commonly used form in order to evaluate that model’s performance. Exploring other formulations of the relationship between LAI and SAVI is good, but it falls outside the scope of this study. In METRIC, the LAI values are derived from SAVI. Maximum values for LAI of 6.0 corresponding to SAVI = 0.69. Beyond SAVI = 0.69 the value from LAI “saturates” and does not change significantly. In METRIC, LAI is limited to 6.0 when SAVI >0.687 and LAI = 0 when SAVI<0.1. (Allen et al. 2007).

Please refer to the attached pdf file which contains the pictures indicates the revision.

Point 9: Figure 8: Please slightly enlarge the font.

Response 9: Corrected:  We modified Figure 8 and enlarged the font

Please refer to the attached pdf file which contains the pictures indicates the revision.

Conclusions:

Point 10: Line 464: The differences in the relationship between LAI and vegetation indices can vary due to reasons beyond spatial scales, such as crop type, structure, phenology, etc. Please refer to (Nguy-Robertson et al, 2012, Agronomy Journal, DOI: https://doi.org/10.2134/agronj2012.0065; Kang et al, 2016. Remote Sensing. DOI: https://doi.org/10.3390/rs8070597) and provide a deeper discussion.

Response 10: Thank you for this comment: We have added additional discussion based on our results

Please refer to the attached pdf file which contains the pictures indicates the revision.

Point 11: Line 482: Again “relationship between LAI, Ts and ETa” is misleading.

Response 11: Corrected: We have revised the sentence as shown below.

Please refer to the attached pdf file which contains the pictures indicates the revision.

Round  2

Reviewer 1 Report

The authors did a remote sensing experiment following the common practice to estimate and validate LAI, Ts and ETa. In a previous round, I recommended a rejection because of its lack of innovation. In the revised version, authors simply replied my minor comments while the manuscript has not been substantially improved. I cannot recommend it for publication.

The reason why we write scientific paper is because we want to solve some scientific questions. Authors need to think what scientific questions they want to address through this study. What’s new of your study compared to others in the literature?

Author Response

Response to Reviewer 1 Comments (Round 2)

Point 1: The authors did a remote sensing experiment following the common practice to estimate and validate LAI, Ts and ETa. In a previous round, I recommended a rejection because of its lack of innovation. In the revised version, authors simply replied my minor comments while the manuscript has not been substantially improved. I cannot recommend it for publication.

The reason why we write scientific paper is because we want to solve some scientific questions. Authors need to think what scientific questions they want to address through this study. What´s new of your study compared to others in the literature?

Response 1: We really appreciate this comment: We have added and rewritten the introduction section to make a better case for why this research is relevant and has scientific merit.
